# Uncovering the subtype-specific temporal order of cancer pathway dysregulation

**Sahand Khakabimamaghani** [ID]*, **Dujian Ding** [ID], **Oliver Snow** [ID], **Martin Ester**

School of Computing Science, Simon Fraser University, Burnaby, British Columbia, Canada

* sahandk@sfu.ca

**Data Availability Statement:** The COAD and GBM data used in this research are respectively provided in DOI: 10.1038/nature11252 and DOI:10.1038/nature07385 and are publicly available from cBioPortal (http://www.cbioportal.org/).

## Abstract

Cancer is driven by genetic mutations that dysregulate pathways important for proper cell function. Therefore, discovering these cancer pathways and their dysregulation order is key to understanding and treating cancer. However, the heterogeneity of mutations between different individuals makes this challenging and requires that cancer progression is studied in a subtype-specific way. To address this challenge, we provide a mathematical model, called Subtype-specific Pathway Linear Progression Model (SPM), that simultaneously captures cancer subtypes and pathways and order of dysregulation of the pathways within each subtype. Experiments with synthetic data indicate the robustness of SPM to problem specifics including noise compared to an existing method. Moreover, experimental results on glioblastoma multiforme and colorectal adenocarcinoma show the consistency of SPM's results with the existing knowledge and its superiority to an existing method in certain cases. The implementation of our method is available at https://github.com/Dalton386/SPM.

## Author summary

Different biological processes within a cell are performed through biological pathways. A biological pathway consists of a group of proteins and other molecules and complex interactions between them. It is known that cancer arises due to malfunction, also known as dysregulation, of one or more pathways. Interestingly, a dysregulation in a patient is often caused by mutations in only one (and not more) molecule in the pathway. This phenomenon is known as mutual exclusivity of mutations and can be used for identification of groups of genes forming (cancer) pathways. The same type of cancer in different patients can result due to different trajectories of dysregulations in possibly different pathways resulting in cancer heterogeneity. Cancer heterogeneity implies that cancer treatment should be personalized according to each patient's specific characteristics and mutations. Therefore, grouping patients based on their pathway dysregulation trajectories into cancer subtypes can help identify different cancer mechanisms, inform subtype-specific treatment strategies and improve efficacy. In this paper, we provide a method that uses patients' mutation information captured by DNA sequencing and identifies dysregulated pathways (i.e. molecules involved in each cancer pathway), cancer subtypes (i.e. groups of patients sharing a common pathway dysregulation trajectory) and subtype-specific pathway dysregulation orders (i.e. trajectories defining the different subtypes). The results on

**Funding:** The authors received no specific funding for this work.

**Competing interests:** The authors have declared that no competing interests exist.

synthetic and real-world data indicate that the method can recover meaningful information about the progression of cancer in different groups of patients.

This is a *PLOS Computational Biology* Methods paper.

## Introduction

It is well understood that human tumors develop over an extended time period through the accumulation of genetic mutations. Thanks to the advancements in DNA sequencing technologies, we now have an increasing amount of high quality genomic data for studying the trajectories of cancer mutations (e.g. The Cancer Genome Atlas (TCGA) [1] and International Cancer Genome Consortium (ICGC) [2]). Two intriguing questions for these studies are: (1) Which mutations are crucial for tumor development, and which are not? In other words, what are the so called "driver" mutations and which mutations are only "passengers"? (2) Is there any meaningful temporal pattern for the driver mutations and how can we infer it if it exists? While the first problem can be solved in part by comparing observed frequencies of mutations across different patients [3], the second problem concerning the temporal order of mutations remains challenging. This is because most of the existing genomic datasets contain cross-sectional data taken from single samples across different individuals and not longitudinal data. Moreover, solving the second problem can provide insights into the relative importance of the driver mutations (i.e. the earlier the mutation time, the more important the mutation).

A number of studies have focused on inferring the temporal order of single mutations in different cancers. These works can be categorized into two main classes. The first class consists of methods based on sequencing data from a single individual. Some of the earlier methods in this class used bulk sequencing data from a single sample (*e.g.* rec-BTP [4], CTPsingle [5]) or multiple samples from the same tumor or patient (*e.g.* PhyloWGS [6], AncesTree [7], LICHeE [8], CITUP [9]). The latest developments are based on single-cell data (*e.g.* OncoNEM [10], SCITE [11], SiFit [12]) or simultaneously utilize single-cell and bulk sequencing to create synergy between the two data types (B-SCITE [13] and PhISCS [14]).

The second class contains methods based on sequencing data from several individuals. Unlike the methods of the first class, ensemble level methods mostly use binarized cross-sectional mutation data indicating whether the mutation has occurred or not. The first work in this class models the relationship between the mutations as a linear path [15]. This was followed by the idea of a phylogenetic tree among the mutations (e.g. [16, 17]). Other works consider different possible evolutionary trajectories for subsets of individuals and infer a mixture of trees (e.g. [18–21]). Others have used a probabilistic graphical model with mutations as the variables to allow for later convergence of mutations (e.g. [22–28]), which is not possible in a tree model. A recent publication using an ensemble approach [29] leveraged non-binary cancer cell fraction data which provides more information about the timing of mutations.

Because the above methods work at the level of single mutation or gene, they are challenged by the extensive heterogeneity in the mutation data, which results in weak temporal patterns. This heterogeneity is associated with the fact that somatic mutations target specific biological pathways [30]. "A biological pathway is a series of actions among molecules in a cell that leads

to a certain product or a change in the cell. Such a pathway can trigger the assembly of new molecules, such as a fat or protein. Pathways can also turn genes on and off, or spur a cell to move". [31] Pathways can be dysregulated due to the mutation of any of their critical member genes. This results in different individuals having mutations on different genes of the same pathway. Based on this observation, cancer is modulated by dysregulation of pathways and the relevant order of mutations might be at the level of pathways not genes [3]. This approach decreases the challenges associated with the heterogeneity of mutations across individuals.

Some previous work has been done on the detection of temporal patterns in pathway dysregulations. Some of these approaches (e.g. [32, 33]), which demonstrate advantages over gene-based methods, are based on the known pathways (e.g. KEGG [34]). Since most annotated pathways are large and overlap with other pathways, they are inapplicable for the discoveries of mutation progression in smaller sets of interacting genes [3]. One alternative is to discover pathways *de novo*. Raphael and Vandin [3] proposed a method to simultaneously infer pathways and the timing relation between them. They hypothesized that tumor development is driven by a linear sequence of pathway dysregulations and proposed the Pathway Linear Progression Model (PLPM). PLPM exploits a property known as mutual exclusivity, which is first mentioned in [35] and indicates that an individual is unlikely to have more than one mutation in a particular pathway. Mutual exclusivity has also been used to successfully identify pathways in cancer datasets [36–39]. To find the optimal PLPM model for the observed cross-sectional sequencing data represented in a binary format, an integer linear program is used. Later, Cristea *et al.* [40] relaxed the linear progression assumption of PLPM in their method called path-TiMEx. They provided a probabilistic graphical model that captures mutual exclusivity as well as the complex progression models. It generalizes both TiMEx [41], which finds mutually exclusive sets of genes, and Conjunctive Bayesian Networks [23, 26], which identifies partial orders of mutations.

However, the effectiveness of PLPM and pathTiMEx is limited because of their simplifying assumptions. Firstly, they assume that all individuals follow the same progression model of pathway dysregulation, but different cancer subtypes have been widely observed in cancer patients [42, 43]. Within the context of cancer analysis, subtypes refer to "the smaller groups that a type of cancer can be divided into, based on certain characteristics of the cancer cells" [44]. Some recent biological studies (e.g. [45, 46]) have indicated a relationship between differences in observed phenotypes and differences in temporal orders of mutations, which implies the possible subtype-specificity of the temporal pattern of mutations. Hence, by assuming a single progression model for all individuals, subtype-specific information from input profiles is lost in the PLPM approach and the risk of making biased inference increases. Secondly, PLPM and pathTiMEx assume binary mutation profiles and cannot exploit the additional information that sequencing technologies (e.g. bulk sequencing) can provide about the temporal order of mutations (e.g. Cancer Cell Fractions (CCFs) extracted from read count data).

In this work, we also assume linear progression among pathways and irreversible mutations (a mutation is inherited by all descendent cells after it occurs in a cell) as in PLPM. To address the aforementioned drawbacks, we relax other assumptions of the PLPM model and propose a novel model, called the Subtype-specific Pathway Linear Progression Model (SPM), that (1) detects different cancer subtypes and their corresponding progression models and (2) exploits CCFs extracted from read count data. In our method, cancer subtypes are captured by the notion of tumor lineage. That is, the subtype of a tumor is identified by clustering the pathway dysregulation orders of individual tumors. We also use the concept of mutual exclusivity for detecting the pathways. However, we incorporate non-binary data providing more information for inference of pathway dysregulation order and allowing for stronger enforcement of mutual exclusivity between more important mutations, i.e. mutations that appear earlier. Last,

but not least, our model can employ both binary and continuous sequencing data, which increases the versatility and accuracy of the model.

The rest of the paper is organized as follows. First, we define the problem of finding the optimal SPM from mutation profiles. Then, an Integer Linear Program solution is provided. Finally, the feasibility and accuracy of the proposed method is demonstrated through experiments with two real cancer datasets.

## Problem definition

In a SPM problem, we are given:

- An $m \times n$ mutation matrix $C$, with samples on the rows and genes on the columns, where $C_{i,j}$ is the Cancer Cell Fraction (CCF) of gene $j$ in sample $i$. CCF of a variant denotes the proportion of cancer cells in a sample that harbour that variant. For a single nucleotide variation (SNV) locus with neutral copy number, the CCF is computed as $\frac{2v}{r+v}$, where $v$ and $r$ are respectively the number of variant and reference read counts mapped to an area covering that locus. This computation is assuming a sample purity of 1, otherwise the read counts should be corrected based on the known purity by reducing the value of $r$. If the sample purity values are not known, CCF values can be replaced with Variant Allele Frequency values computed as $\frac{v}{v+r}$. For loci with copy number variation (CNV), computation of CCF is very difficult and associated with uncertainty. Therefore, we do not consider loci with CNV regardless of occurrence of SNV in the locus. For a gene, we take the largest CCF of the corresponding variants if there are more than one variant in that gene. Based on our assumptions, i.e. linear progression and irreversible mutations, this choice refers to the earliest mutation that has happened for that gene.

- An integer $K$, which indicates the number of pathways to discover.

- An integer $T$, which indicates the number of subtypes to discover.

- A tolerance value $\epsilon$, which indicates the sequencing noise. If the difference between two CCFs is smaller than $\epsilon$, they will be accounted the same. Larger $\epsilon$ should be used for lower sequencing coverage.

We are interested in the following outputs:

- A partitioning of genes into $K$ pathways, such that mutual exclusivity is maximized within each pathway. Specifically, we want to avoid having pathways that include two or more simultaneous mutations with large CCFs, i.e. the larger the CCFs, the larger the penalty because the larger CCFs indicate generally stronger signals and more confident mutations.

- A partitioning of samples into $T$ subtypes, such that the pathway linear progression is consistent among samples of each subtype as described next.

- For each subtype, a linear progression model capturing the linear temporal order of dysregulation of the $K$ pathways. In other words, linear progression corresponds to a sequence of pathways with decreasing CCFs with a tolerance $\epsilon$. Thus, pathway $x$ is *necessarily* dysregulated before pathway $y$ in sample $i$ if and only if $Z_{i,x} - Z_{i,y} > \epsilon$. Otherwise, if $|Z_{i,x} - Z_{i,y}| \leq \epsilon$, the order of the two pathways is ambiguous for sample $i$ and should be identified based on the observations for other samples in the same subtype. We note that the CCF value is considered correlated with mutation timing based on the assumptions of linear cancer progressions and irreversible mutations and excluding the CNVs. Under a different set of assumptions, CCF and mutation time are not necessarily correlated.

Inputs:

Hyper-parameters

$K = 3, T = 3, \epsilon = 0.1$

Matrix $C$

**Fig 1. An example input with the corresponding outputs.**

Outputs:

Pathways

$P_1 = \{g_1, g_2, g_3\}$

$P_2 = \{g_4, g_5\}$

$P_3 = \{g_6, g_7\}$

Fig 1 shows an example of inputs (left) and outputs (right). One possible SPM is color-coded in the input matrix and also depicted in the output. Subtypes in the solution are identified by different colors in the CCF matrix. The intensity of colors is correlated with the CCF values. The CCF values corresponding to each pathway are shown in bold and the false positive detected is underlined. Because the progression order in subtype 3 is $P_3 \rightarrow P_1$, the value of largest CCF for sample $r_7$ in pathway $P_1$ is 0.4 (for $g_1$), and $0.4 - 0.2 > \epsilon = 0.1$, this largest CCF is not selected for pathway $P_1$ in sample $r_7$ (i.e. is detected as false positive) and the value for $g_3$ is instead chosen as the corresponding CCF. The white circles indicate the initial normal state. The pathway progression steps are shown by colored circles with the connecting edges labeled by dysregulated pathways.

**Theorem 1**. *The SPM problem is NP-hard for any $K \geq 2$, $T \geq 1$, and $\epsilon \geq 0$.*

*Proof.* This proof is inspired by the proof provided in [47] for NP-hardness of PLPM problem. We extend that proof to incorporate CCF values instead of binary mutation data and generalize it to the case of having multiple subtypes.

We prove by reduction from weighted *Minimum-UnCut* problem [48], which is a well-known NP-Complete problem. The input to the weighted *Minimum-UnCut* problem is a graph $G = (V, E)$ where each edge $e \in E$ has a positive weight $w_e$. The output is a cut in $G$ resulting in two sets $S$ and its complement $\bar{S}$ such that the weight sum of edges $\{e_{ab} : a, b \in S \ \lor \ a, b \in \bar{S}\}$ (i.e. edges within the two sets) is minimized.

Given a graph $G$ and the edge weights $w_e$, we construct the equivalent SPM problem as follows. First, we assume $K \geq 2$ pathways and genes $\{g_1, \ldots, g_n\} \cup \{g_{n+1}, \ldots, g_{n+K-1}\}$, where $n = |V|$ and $\{g_{n+1}, \ldots, g_{n+K-1}\}$ are auxiliary genes. Second, we assume $T \geq 1$ subtypes and assign a pathway progression order for each subtype. Pathways $K$ and $K - 1$ get positions 1 and 2 respectively for all subtypes. The position of pathways $l \leq K - 2$ are selected arbitrarily and the corresponding position of pathway $l$ in the progression order for subtype $t$ is indicated as $Pos_t(l)$, which is between 3 and $K$.

Now we construct the input matrix $C$ for the SPM problem as follows. For every $e = (u, v) \in E$, $T$ samples $r_{1,t}^e$ ($1 \leq t \leq T$) are generated such that for each $t$, we set $g_u = g_v = w_e + \Delta$ ($\Delta > \epsilon$) in the corresponding sample. Additional $\Delta$ is to make sure that the weights will be larger than

the tolerance $\epsilon$. The values for genes in $\{g_1, \ldots, g_n\}$ other than $u$ and $v$ are set to 0. Let $\hat{w}$ be the maximum edge weight for $G$. Then, for sample $r^e_{1,t}$ we set $g_{n+l} = \hat{w} + Pos_t(l) \times \Delta$ for $\{l: Pos_t(l) > 1\}$. Similarly, we define $T$ samples $r^e_{2,t}$ for every $e \in E$, except that we set values of auxiliary genes as $g_{n+l} = \hat{w} + Pos_t(l) \times \Delta$ for $\{l: Pos_t(l) > 2\}$. So the auxiliary gene $g_{n+K-1}$ is set to 0.

This results in a matrix $C$ with $2T|E|$ rows and $n + K - 1$ columns. Since we want to maximize the mutual exclusivity of mutations of each sample $s$, if there are more than one mutated genes assigned to a pathway, all but one mutation will be considered as false positive and the cost for that assignment will be equal to sum of the CCF values of false positive mutations. The choice of the true positive mutation among those mutations within the pathway depends on both the CCF values of those mutations (smaller CCFs are most likely to be false positive) and the progression constraint (based on the CCFs of the pathways dysregulated before and after that pathway). We indicate the total cost for a sample $s$ as a result of a pathway assignment $\rho$: $\{1..n\} \rightarrow \{1..K\}$ by $f_\rho(s)$. For each $e$ and $t$ there are different possibilities in a solution $\rho$ for SPM problem. Denoting the pathways of genes $u$ and $v$ as $\rho(u) = l_u$ and $\rho(v) = l_v$, there are following possibilities for each subtype $t$:

1. $Pos_t(l_u), Pos_t(l_v) > 2$ (both genes are outside of the pathways $K$ and $K - 1$): Both genes will be considered as false positive as there is a larger value for $g_{n+l_u}$ and $g_{n+l_v}$ within the pathways $l_u$ and $l_v$. So, for all $e \in E, f_\rho(r^e_{1,t}) = f_\rho(r^e_{2,t}) = 2w_e$ and $f_\rho(r^e_{1,t}) + f_\rho(r^e_{2,t}) = 4w_e$.

2. $Pos_t(l_u) \leq 2$ and $Pos_t(l_v) > 2$ (one gene is inside one of the pathways $K$ of $K - 1$ and other gene is outside them): Assuming that $Pos_t(K) = 1$, if the gene $u$ is inside pathway $K$, then, for all $e \in E, f_\rho(r^e_{1,t}) = w_e$ and $f_\rho(r^e_{2,t}) = 2w_e$. Else, if $u$ is inside pathway $K - 1$, then $f_\rho(r^e_{1,t}) = 2w_e$ and $f_\rho(r^e_{2,t}) = w_e$. So, in any case, $f_\rho(r^e_{1,t}) + f_\rho(r^e_{2,t}) = 3w_e$. This is also true for $Pos_t(K) = 2$.

3. $Pos_t(l_u) = Pos_t(l_v) \leq 2$ (both genes are in one of the pathways $K$ or $K - 1$): In either case when both $u$ and $v$ are in $K$ or in $K - 1$, we have $f_\rho(r^e_{1,t}) + f_\rho(r^e_{2,t}) = 3w_e$ for all $e \in E$.

4. $Pos_t(l_u), Pos_t(l_v) \leq 2$ and $Pos_t(l_u) \neq Pos_t(l_v)$ (both genes are in one of the pathways $K$ or $K - 1$, but not in the same pathway): We have $f_\rho(r^e_{1,t}) + f_\rho(r^e_{2,t}) = w_e$ for all $e \in E$.

Based on the weighting scheme and the costs of the four mentioned cases, it is easy to see that in the optimal pathway assignment, (1) all genes $\{g_1, \ldots, g_n\}$ will be in pathways $K - 1$ and $K$, (2) each auxiliary gene in $\{g_{n+1}, \ldots, g_{n+K-2}\}$ will be in a separate pathway, (3) $g_{n+K-1}$ will be assigned to pathway $K - 1$, and (4) the order of pathway progression for each subtype will follow the arbitrarily assigned orders. Accordingly, we will have only costs shown in cases 3 and 4 above. Thus, the total cost of the optimal solution will be $3\lambda_3 + \lambda_4 = TW + 2\lambda_3$, where $\lambda_3$ and $\lambda_4$ are the sum of weights of the edges corresponding to samples of cases 3 and 4, respectively, and $W = \Sigma_{e \in E} w_e$. Since $TW$ is constant, the optimal solution minimizes the $\lambda_3$ value which is $T$ times the objective for the weighted *Minimum-UnCut* problem. Indeed, the two pathways $K - 1$ and $K$ correspond to the sets $S$ and $\bar{S}$ of *Maximum-Cut/Minimum-UnCut* in graph $G$. This concludes that SPM problem is at least as complex as the *Minimum-UnCut* problem.

As the last piece of the proof, we show that reduction from weighted *Minimum-UnCut* problem can be performed in polynomial time. The equivalent SPM input matrix $C$ constructed as above for a weighted *Minimum-UnCut* problem on a graph $G = (V, E)$ contains $2T|E|$ rows, with a cost of $O(n + K)$ for each row. This results in a total polynomial complexity of $O(T|E|(n + K))$. The cost of generating the arbitrary subtype-specific progression orders $Pos$ is $O(TK)$ and does not change the overall big-oh cost. This concludes that the SPM problem is NP-hard.

## Simulation results

### Synthetic data generation

Different synthetic datasets are generated based on different configurations of four factors: $m$, $T$, $K$ and CCF noise. For a given configuration, first $m$ patients are assigned to $T$ subtypes with a uniform distribution. Then, genes are randomly assigned to pathways based on input $K$ assuming equal size for all pathways. For each subtype, a unique random linear progression order is generated. Then, for each patient, $K$ genes are randomly selected from $K$ different pathways. CCF values of these genes are set according to the progression order using a Dirichlet distribution for the distances between the values of consecutive CCFs, assuming value of 1 for the largest CCF. Then, a noise of $\lambda \in \{0.01, 0.05, 0.1\}$ is added to the input matrix in two steps: 1) adding a Gaussian noise with standard deviation of $\lambda$ to the non-zero elements of the current matrix and 2) by selecting $mn\lambda$ of the zero elements of the input matrix and changing their value to $2\lambda$. The second step introduces false positives and the value of $2\lambda$ is used to ensure that these values will not be detected by a filter that removes mutations with a small CCF.

### Sensitivity analysis

The tested values for the four factors considered in the simulations and their default settings are shown in Table 1. When analysing the sensitivity to a factor, the other factors are set to their default values. In addition to these factors, we consider running the algorithm with different tolerance values $\epsilon \in \{0.05, 0.1, 0.2\}$. We limited the running time to 10 hours and available memory to 128 GB for both methods.

The following measures are used for performance evaluation. The accuracy of identified subtypes and pathways is measured by the Rand index between the ground-truth clusters and the methods' outputs. With respect to detecting the correct progression order, we used 1–Kendall tau distance to measure the similarity between the ground-truth orders and methods' outputs. Because there are multiple orders for different subtypes, we first associate each output subtype and pathway to its corresponding most overlapping ground-truth subtype and pathway. For SPM, for each identified subtype, we compute the Kendall tau distance with the order of the corresponding ground-truth subtype and take the average over all subtypes. For, PLPM, because it does not produce subtypes, we report the value for the ground-truth order with the maximum similarity. In cases with more than one solutions, we take the mean of the mentioned measures over all solutions.

The results are shown in Fig 2. As expected, because PLPM does not detect subtypes, the Rand index of its outputs is always smaller than SPM. SPM performs an almost perfect subtyping and pathway detection in most of the experiments. SPM outperforms PLPM in detecting pathways because of its ability to consider multiple different progression orders for different subtypes. Moreover, in presence of higher noise, binary data results in worse performance for PLPM as it assigns similar weights to small (uncertain) and large (certain) CCFs. The pathways

**Table 1. Factors of synthetic data generation and their values.** The default values are underlined.

| Factor | Value |
|---|---|
| $m$ | 50, <u>100</u>, 200 |
| $K$ | 3, <u>4</u>, 5 |
| $T$ | 2, <u>3</u>, 4 |
| Noise | 0.01, <u>0.05</u>, 1 |

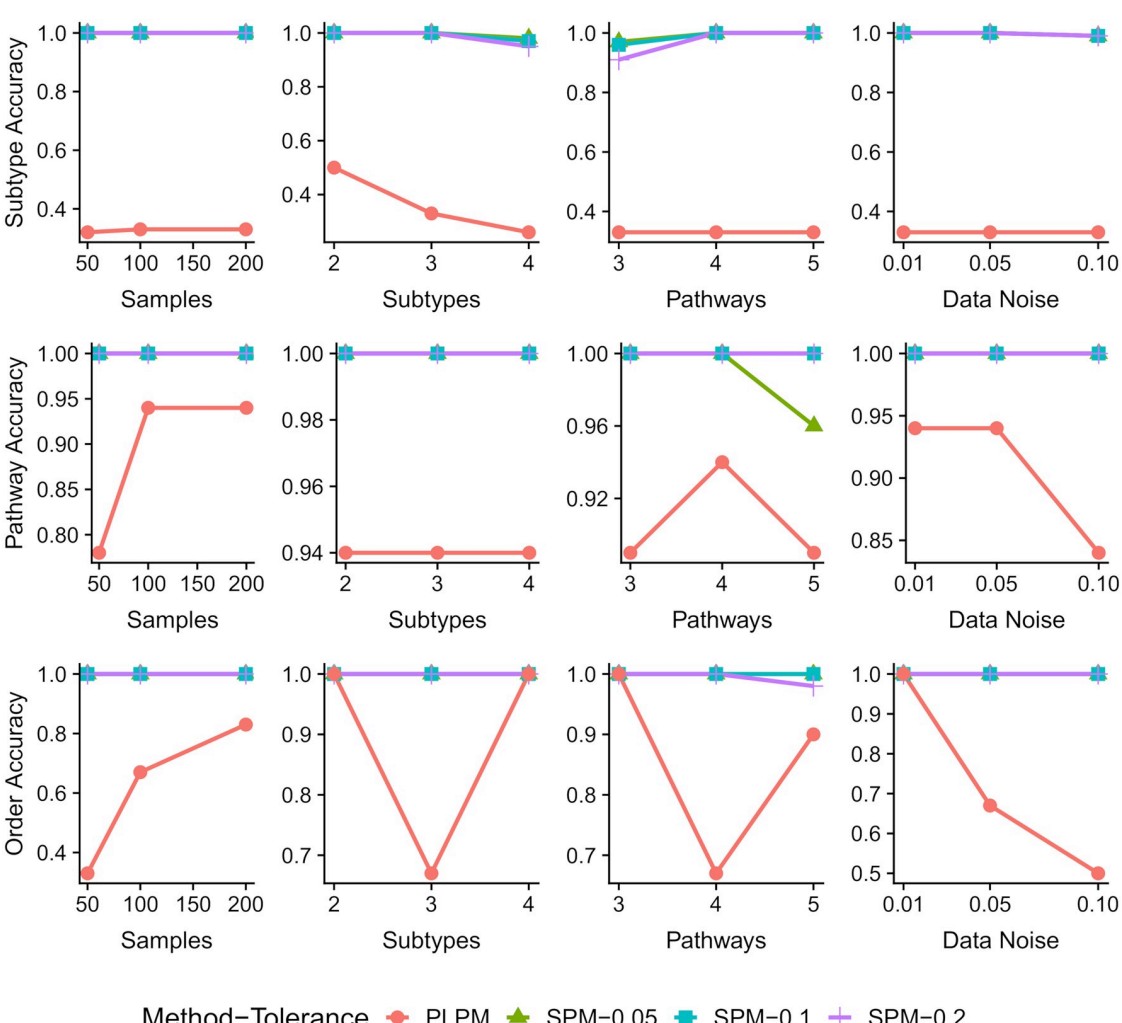

**Fig 2. Results of sensitivity analysis using synthetic data.** Note that each graphs might have different scales for the vertical axes.

and their order detected by PLPM are also more sensitive to the number of provided samples *m* and PLPM require more samples to detect the correct pathways and an order matching progression of one of the subtypes. PLPM can detect one of the subtype progression orders in three cases: 1) with 2 subtypes, because, compared to 3 subtypes, there are more samples per subtype and less number of conflicting progression orders, 2) 4 subtypes, because the particular random example generated for this case had very similar progression orders across different subtypes and 3) low noise, because of small number of false positives. However, PLPM does not assign the genes to the pathways as accurate as SPM does in those cases.

## Real data results

### Datasets

We evaluated our method by applying it to two datasets from TCGA containing somatic mutation data from patients with colorectal adenocarcinoma (COAD) [43] and glioblastoma multiforme (GBM) [42]. As described earlier, we used CCF values as opposed to the binarized data used in previous works. This enables us infer more about the proper ordering of mutations but

limited us to only somatic mutation information as copy number variation data would reduce the accuracy of the analysis. Although there are tools for inferring the CCF values for copy number variations (e.g. PyClone [49]), their accuracy is limited for low-coverage cross-sectional data, including the two datasets we investigated. Therefore, we opted to filter out any sample that had a copy number variation in any of genes of the panels we studied. This filtering avoids the type of bias that can be caused by ignoring CNVs that indeed exist in a patient. Although this process reduces sample size and limits our biological conclusions to the relationships between SNVs, the implications are still interesting with respect to the cancer progression and involved mutations.

Moreover, all weak signals including the loci with total read count less than 20, variant read count less than 10, or CCF smaller than 0.05 were considered false positives and were eliminated. To tolerate the potential data noise, we set $\epsilon = 0.2$ for SPM. This value was selected from values {0.05, 0.1, 0.2} based on the consistency of the output pathways and subtypes with the existing knowledge about the pathways or using clinical data (e.g. survival or disease stage).

We focused our analysis on the list of driver genes considered in [47] for evaluating PLPM. After the mentioned filtering of CNVs and weak signals, the COAD dataset included 132 samples and 14 genes (*APC*, *FBXW7*, *ACVR2A*, *AMER1*, *PIK3CA*, *TCF7L2*, *TP53*, *BRAF*, *KRAS*, *NRAS*, *SMAD2*, *SMAD4*, *SOX9*, *ELF3*) with at least one mutation in those samples. The GBM dataset consisted of 69 samples and 15 genes (*IDH1*, *PTEN*, *FGFR1*, *EGFR*, *NF1*, *RB1*, *TP53*, *PIK3R1*, *PIK3R2*, *PIK3CA*, *PIK3CB*, *PIK3C2G*, *FGFR3*, *PIK3C2A*, *ATRX*). 12 genes out of 27 initial genes used in [47] were filtered out in our preprocessing as they corresponded to copy number variations. To compare the results of SPM and PLPM for the same datasets, we binarized these COAD and GBM datasets and applied PLPM on them.

As per the hyper-parameters, we tried the range of values $T \in \{1, \ldots, 4\}$ and $K \in \{2, \ldots, 8\}$ and picked the result with the best value of the objective function. Each experiment was performed in a parallel mode with 32 CPU cores using Rcplex R package [50], which is an R interface for CPLEX solver [51]. We limited the running time and memory to at most 40 hours and 256 GB. Due to the complexity of the model and small number of samples, some of the experiments reached these limits and the solutions were not guaranteed to be optimal. However, the optimality gaps computed by CPLEX for the presented results, which were respectively ∼13% and ∼0% for COAD and GBM datasets, indicate that the solutions are near optimal. As the sequencing technologies are advancing, the increasing quality of data will make it possible to incorporate copy number variation data and more samples, which is expected to inform the searching strategy to find better solutions in shorter time.

## Glioblastoma multiforme

GBM has been well studied and a number of notable genes and pathways have been implicated in the progression of the disease [52]. Mutations in the canonical cancer genes *TP53* and *EGFR* are well known drivers of GBM in addition to disruptions in the PI3K pathway. Applied to the GBM dataset, SPM identified four distinct pathways that were significantly altered in samples, recapitulating much of the known pathways involved in GBM (Fig 3). In three out of four cases, SPM groups together two or more genes from a known pathway and is reasonably good at separating pathways from each other. Where SPM struggles is in grouping the PI3K pathway genes together, spreading them across identified pathways. Despite this, SPM excels in preserving the mutual exclusivity of the pathways, with no violations of this property in any of the cases. This is due to the freedom of having multiple subtypes. In contrast, PLPM violates the mutual exclusivity property in 10 cases, equating to ∼14%, for the first pathway (Fig 3).

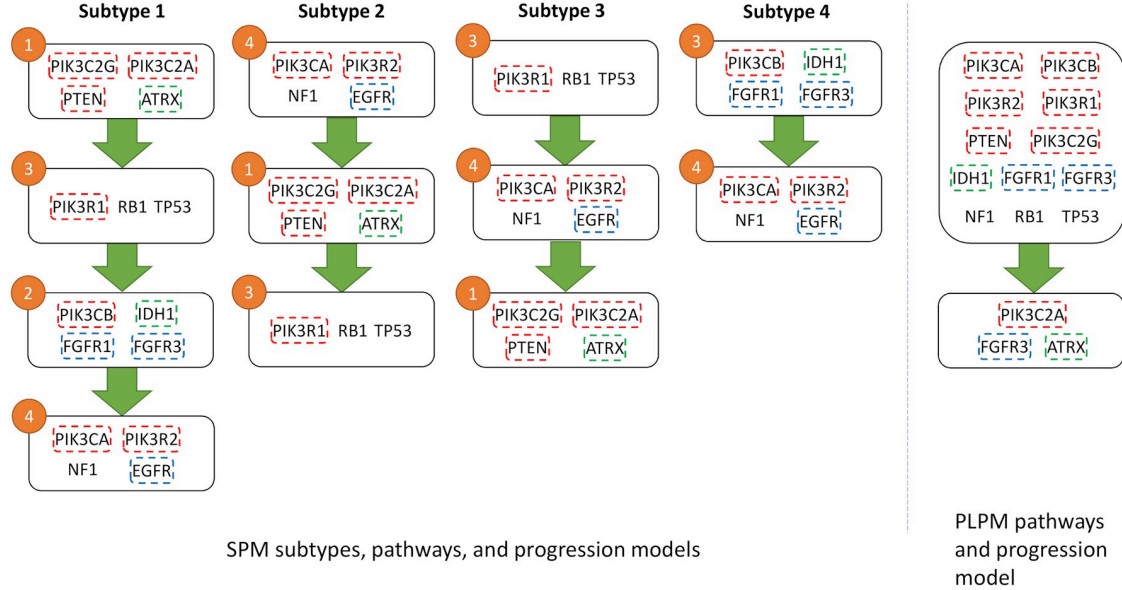

**Fig 3. Results for GBM using SPM (left) and PLPM (right).** Genes within the same known pathway are coded by the same color. The numbers on pathways refer to the pathway numbers in Fig 5.

The reason for this is that the first pathway that PLPM identifies contains 12 genes that correspond to multiple (at least four) known biological pathways. Thus it is highly probably that two genes from this pathway will be altered in the same sample, therefore violating the mutual exclusivity property.

Four unique GBM subtypes were also discovered, in contrast to the single subtype of PLPM, based on the different temporal ordering of mutations in the four pathways. Interestingly, only the first subtype contains aberrations in all of the four identified pathways, while the subtypes 2, 3, and 4 have 3, 3, and 2 mutated pathways respectively. Survival analysis (Fig 4) shows subtype 4 having noticeably better survival probability than the other subtypes, suggesting that the temporal order of mutations is predictive of cancer progression. Importantly, subtype 4 has no alterations in *TP53* or *PTEN*, two of the main drivers of GBM.

Additionally, two of the detected subtypes map reasonably onto two of the five subtypes of GBM described in [52]. Subtype 4 captures the G-CIMP subtype with Jaccard index 0.47 (FDR p-value 0.04) and subtype 2 maps to Neural subtype with Jaccard index 0.33 (FDR p-value 0.07). In particular, both G-CIMP-high and our subtype 4 are characterized by a high frequency of mutations in *IDH1*, and in our case is the first mutation to occur (Fig 5). Survival analysis in [52] shows a striking difference between the G-CIMP-high subtype and their other molecular subtypes, similar to the difference seen in our survival analysis. A recent review [53], reinforces the connection between G-CIMP-high tumours and mutations in *IDH1* and the favourable prognoses associated with this subtype. This further supports the importance of our identified subtypes as they may be predictive of patient outcomes. Furthermore, since the pathway order is directly related to CCF values, earlier mutations in pathways will be more populous in the tumour. The relatively small size of each of our pathways makes this even more actionable from a clinicians standpoint, as opposed to the large and uninformative first pathway from PLPM.

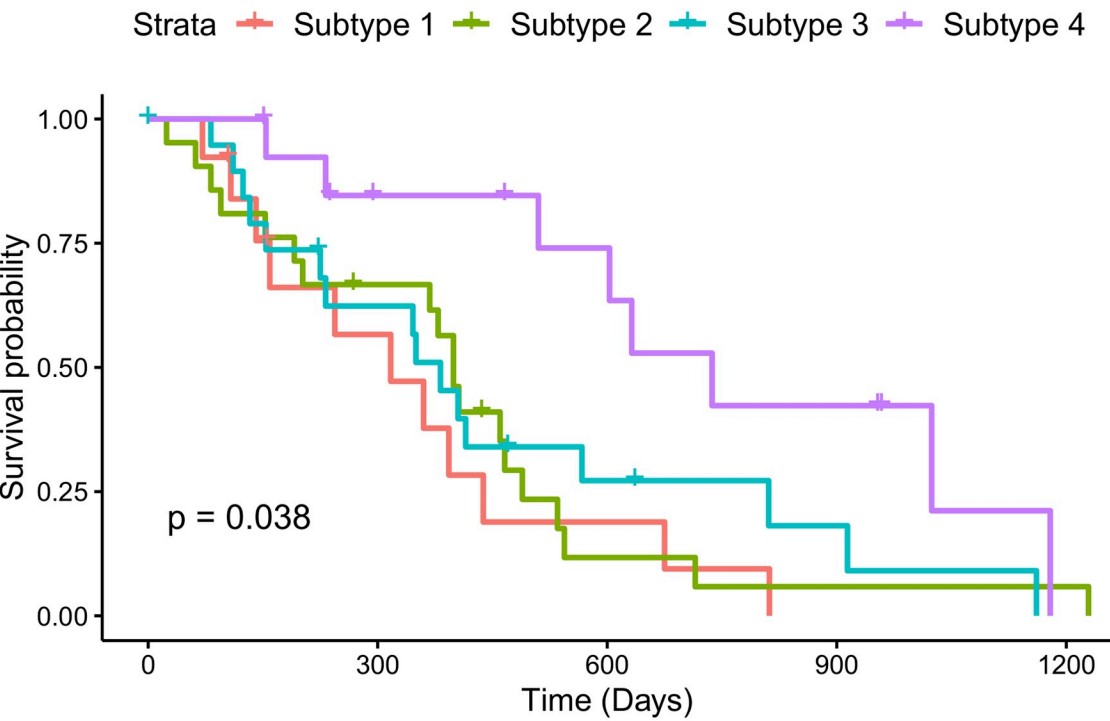

**Fig 4. Survival analysis for identified subtypes of GBM.**

### Colorectal adenocarcinoma

When applied to the COAD dataset, SPM again identifies four distinct pathways and four subtypes based on the temporal ordering of those pathways (Fig 6). However, in contrast to the GBM results, all pathways are altered in each subtype, with no wild-type pathways present. The separation of genes into their known biological pathways, as defined in [43], is less clear with the COAD dataset, with genes being spread across the identified pathways. However, SPM is still effective at maintaining mutual exclusivity with only an average of 13 cases ($\approx 10\%$) per pathway as opposed to PLPM which violates this property in 22 cases ($\approx 16\%$) per pathway.

Although we could not recognize any advantage for any of the four identified progressions compared to the progression from PLPM and no single progression fully recapitulating the known pathways, they provide a meaningful way to stratify patients by correlating to disease subtype. Therefore, identifying the separate subtypes can still be informative, in particular for finding the proper ordering of mutations. Raphael and Vandin [47] made the argument that *TP53* mutations occur before *KRAS* mutations, in contrast to previous reports for COAD but in line with more recent work on other cancer types. Although their progression supported this hypothesis, it is in conflict with the high CCF values for *KRAS* in some samples (Fig 5). In 88 cases ($\approx 67\%$), the progression from PLPM does not match the ordering of the CCF values, whereas SPM does not suffer from this issue due to its hard constraints. Our resulting subtypes are in agreement with the CCF values and suggest that the ordering of *KRAS* and *TP53* is perhaps more variable than once thought, with *KRAS* preceeding *TP53* in subtypes 1 and 4 and *TP53* preceeding *KRAS* in subtypes 2 and 3. Finding a single progression based on binarized data and not CCF values limits this kind of inquiry.

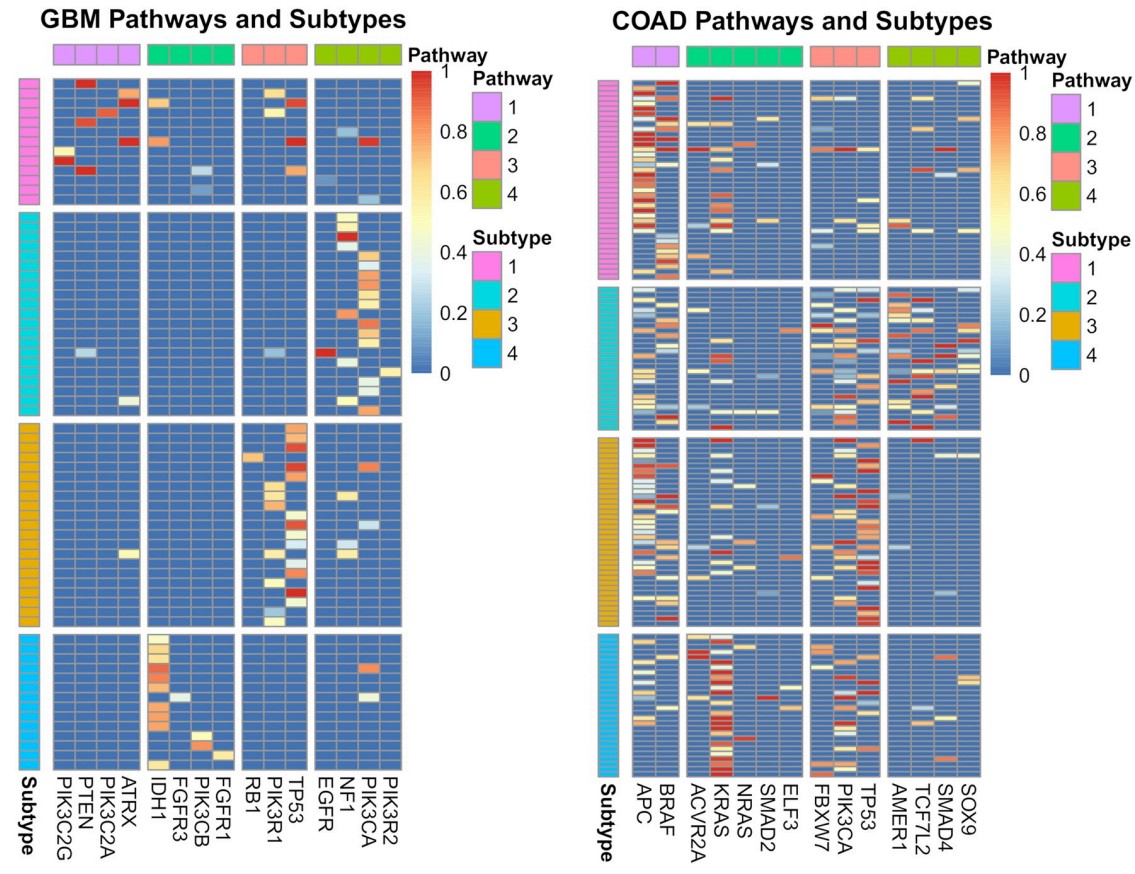

**Fig 5. Heatmap showing CCF values for GBM (left) and COAD (right) datasets partitioned into pathways and subtypes.**

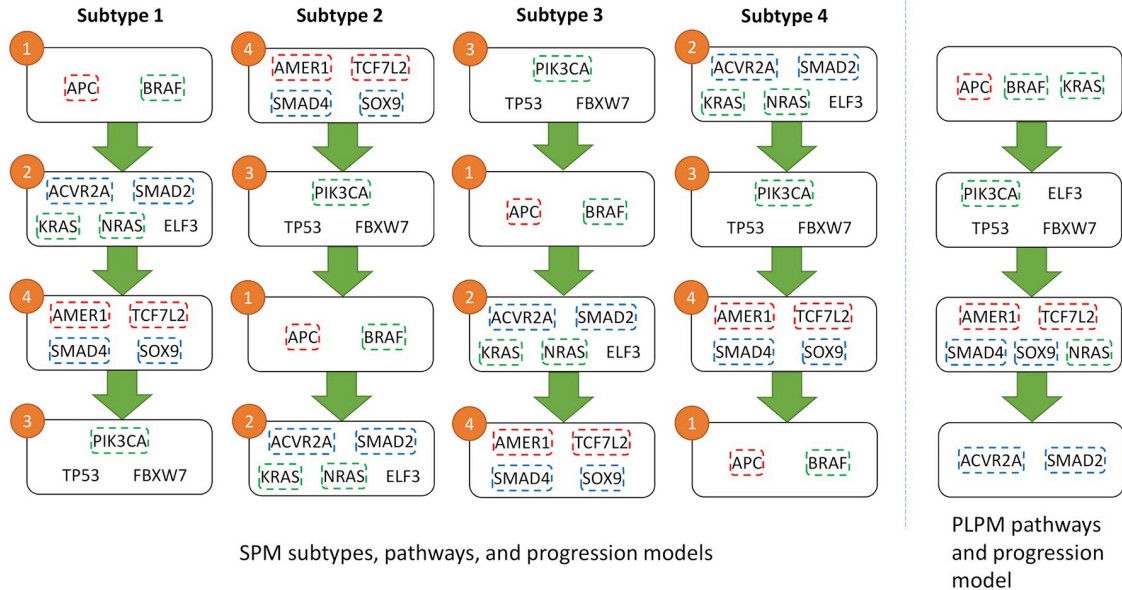

**Fig 6. Pathways detected for COAD using SPM (left) and PLPM (right).** Genes within the same known pathway are coded by the same color. The numbers on pathways refer to the pathway numbers in Fig 5.

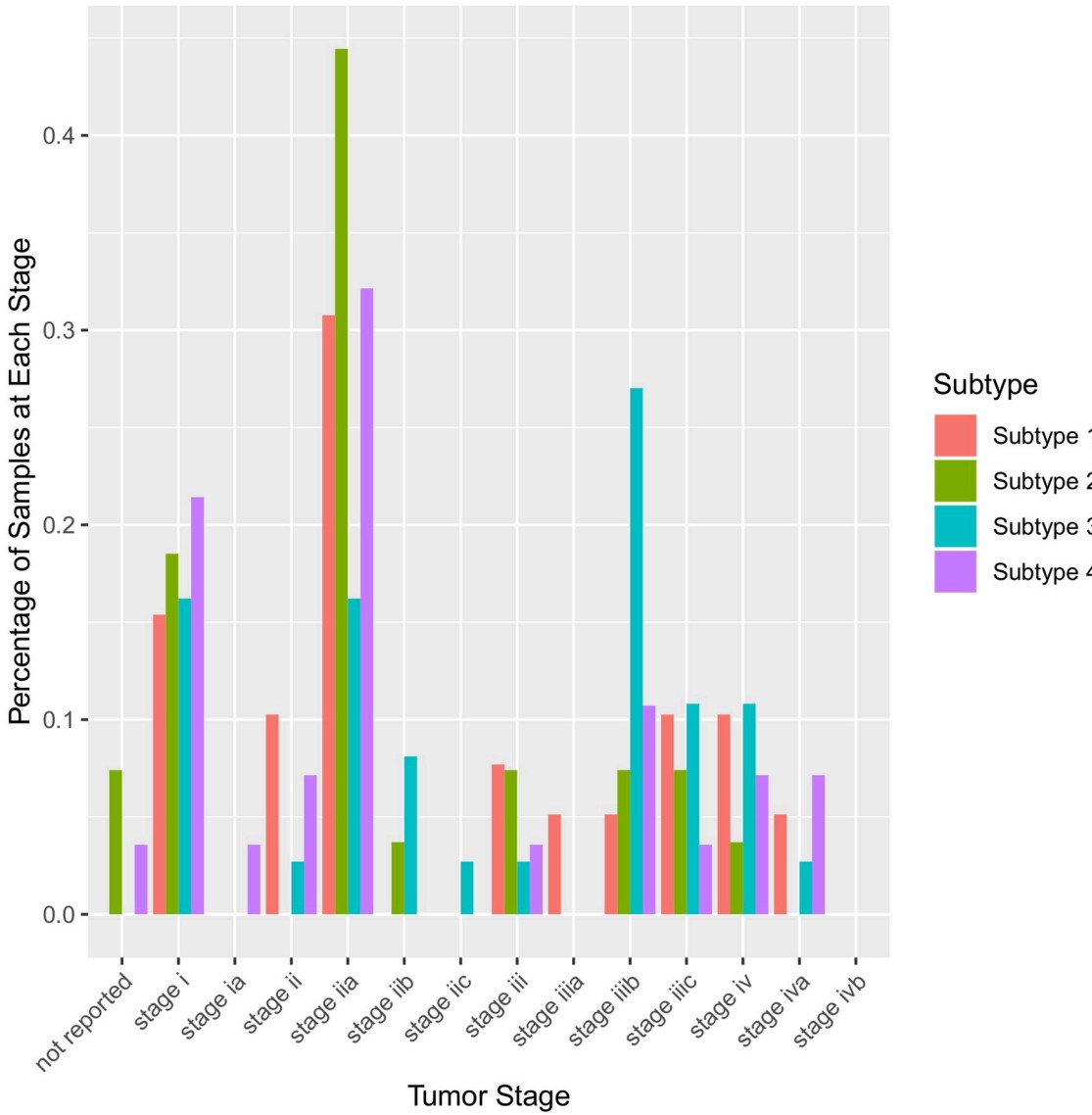

**Fig 7. Prevalence of each identified COAD subtype in different disease stages.**

The resulting subtypes for COAD failed to show any meaningful difference in the survival analysis, unlike the GBM results. However, the subtypes did show some separation when aligned to the tumour stage, a clinical variable included in the COAD dataset (Fig 7). Subtype 3 appears to be more prevalent in stage iiib compared to other subtypes, indicating that subtype 3 maybe be more agressive than the other subtypes. This is further supported by the fact that subtype 3 is characterized by a high frequency of mutations with large CCF values in *TP53*, a major driver in all cancers but especially crucial in COAD progression where *TP53* mutations are associated with drug resistance and poor prognosis [54]. These results suggest that these identified subtypes may be informative for categorizing samples according to tumour stage and predicting patient outcomes.

## Methods

We propose an integer linear program to solve the SPM problem, which simultaneously optimizes pathways, cancer subtypes, and subtype-specific pathway linear progression. Before describing our solution for the SPM reconstruction problem, we explain the existing solution for the PLPM problem proposed in [47], upon which we build the solution of the SPM problem.

### Integer linear program for the PLPM problem

Raphael and Vandin [47] proposed an integer linear programming (ILP) method for the PLPM. Given a binary matrix $M \in \{0, 1\}^{m \times n}$ indicating the existence (1) or absence (0) of a mutation in genes $\{g_1, g_2, \ldots, g_n\}$ for samples $\{r_1, r_2, \ldots, r_m\}$, where rows and columns of $M$ respectively correspond to the samples and the genes, and a parameter $K$ indicating the number of pathways to be discovered, they introduced a model for partitioning the genes into $K$ pathways $P = \{P_1, \ldots, P_K\}$ such that:

- For all $1 \leq k \leq K$ we have $|\{g_j \in P_k : M_{i,j} = 1\}| \leq 1$ (mutual exclusivity)

- For all $1 \leq k \leq K$, if $|\{g_j \in P_k : M_{i,j} = 1\}| > 0$ then $|\{g_j \in P_{k-1} : M_{i,j} = 1\}| > 0$ (progression)

The first constraint guarantees the mutual exclusivity of genes inside each pathway $k$ and the second constraint ensures that if pathway $k$ has a mutation then all preceding pathways in the progression model ($P_{k'}, k' < k$) are also mutated (i.e., have at least one mutation). Then, they define the objective function as the number of elements of the matrix $M$ that need to be flipped ($0 \rightarrow 1$ or $1 \rightarrow 0$) such that the above-mentioned constraints are satisfied. Their objective was:

$$\min \sum_{i=1}^{m} \sum_{k=1}^{K} \sum_{j=1}^{n} (M_{i,j} p_{j,k} - a_{i,k} + 2f_{i,k}),  \tag{1}$$

where $p_{j,k}$ is a binary variable indicating whether gene $g_j$ is assigned to pathway $P_k$ ($p_{j,k} = 1$) or not ($p_{j,k} = 0$), $a_{i,k}$ is a binary variable indicating whether sample $r_i$ has mutation (after the required flips) in pathway $P_k$ ($a_{i,k} = 1$) or not ($a_{i,k} = 0$), and $f_{i,k}$ is a binary variable indicating whether any of the genes in $P_k$ for sample $r_i$ should be flipped $0 \rightarrow 1$ ($f_{i,k} = 1$) or not ($f_{i,k} = 0$). Then the constraints for the linear program are as follows:

- Each gene is assigned to exactly one partition: for $1 \leq j \leq n$, $\sum_{k=1}^{K} p_{j,k} = 1$;

- For each pathway $P_k$, at least one gene is assigned to it: for $1 \leq k \leq K$, $\sum_{j=1}^{n} p_{j,k} \geq 1$.

- PLPM is satisfied for each sample: for $1 \leq i \leq m$ and $1 \leq k \leq K - 1$, $a_{i,k} \geq a_{i,k+1}$.

- For each sample $r_i$, pathway $P_k$ is considered mutated if it has a 1 in $r_i$ or if one of its entries in $r_i$ is flipped to make it mutated (i.e. $f_{i,k} = 1$): for $1 \leq i \leq m$ and $1 \leq k \leq K$, $\sum_{j=1}^{n} M_{i,j} p_{j,k} + f_{i,k} \geq a_{i,k}$.

### Integer linear program for the SPM problem

As stated earlier, we assume that the linear relationships between the pathways are subtype-specific, meaning that the samples can be divided into groups with different linear orders. More specifically, in our model, each subtype is associated with a distinct linear progression model between $K$ pathways.

To model cancer subtypes and continuous CCF values instead of binary mutations, new variables are introduced and some of the variables in PLPM ILP are replaced, resulting in a total of $[Kn(m + 1) + T(m + K^2)] \in O(mnK)$ binary latent values. We add the SPM inputs $C$, $T$, and $\epsilon$ as well as the following variables and inputs:

- A binary variable $s \in \{0, 1\}^{m \times T}$, where $s_{i,t}$ denotes that whether sample $i$ belongs to subtype $t$. The total number of subtypes is restricted by an input parameter $T$, which is given as input. For $1 \le t \le T$ and $1 \le i \le m$, $s_{i,t} = 1$ means that sample $i$ is consistent with the progression in subtype $t$ and $s_{i,t} = 0$ means otherwise.

- A binary variable $A \in \{0, 1\}^{T \times K \times K}$ representing the ancestry matrices of relationships of pathways for each subtype. $A_{t,k,l}$ denotes that, according to the progression model defined by subtype $t$, whether the mutation of pathway $k$ happens before pathway $l$ ($A_{t,k,l} = 1$) or not ($A_{t,k,l} = 0$). $A$ captures and constraints the linear order of mutation of pathways.

- A binary input $M \in \{0, 1\}^{m \times n}$ indicating whether $C_{i,j} > 0$ ($M_{i,j} = 1$) or not ($M_{i,j} = 0$). $M_{i,j}$ is the binarization of the original CCF matrix and facilitates the design of integer linear program as described later.

- A binary variable $\alpha \in \{0, 1\}^{m \times n \times K}$ with $\alpha_{i,j,k}$ indicating whether gene $j$ of sample $i$ corresponds to the CCF value representative of pathway $k$ ($\alpha_{i,j,k} = 1$) or not ($\alpha_{i,j,k} = 0$). If this variable is equal to 1, it means that first, gene $j$ is assigned to pathway $k$ ($p_{j,k} = 1$), second, there is a mutation in gene $j$ of sample $i$ ($M_{i,j} = 1$), and third, considering the CCF of gene $j$ for pathway $k$ in sample $i$ is consistent with the progression model and results in an optimal model. Thus, $\alpha_{i,j,k}$ gives us control over the choice of mutation within each pathway to optimize the objective function.

We also extend the constraints of the PLPM ILP. Firstly, similar to constraints for pathways, we introduce two constraints to enforce a surjective function from samples to subtypes. These constraints will guarantee each subtype is assigned at least one sample and each sample is assigned to exactly one subtype. Second, we introduce a constraint to regulate the behavior of $\alpha_{i,j,k}$, that is, $\alpha_{i,j,k} = 1$ can be true only if $p_{j,k} = 1$. Third, progression constraints are updated using both $A_{t,k,l}$ and $s_{i,t}$ in order to satisfy our assumption of subtype-specificity. Additionally, since bulk-sequencing data become more and more reliable as the sequencing technologies improve and since, unlike binary data, flipping from 0 to 1 is not accurate for continuous CCF data, we assume there is no false negative in the observed mutation matrix.

More formally, the $[m + n + mn + K + T + KT + mK(n + 1 + T(K - 1)/2)] \in O(\max\{mTK^2, mnK\})$ constraints for SPM ILP are as follows:

- Each gene is assigned to exactly one pathway: for $1 \le j \le n$, $\sum_{k=1}^{K} p_{j,k} = 1$.

- For each pathway $P_k$, at least one gene is assigned to it: for $1 \le k \le K$, $\sum_{j=1}^{n} p_{j,k} \ge 1$.

- Each sample is assigned to exactly one subtype: for $1 \le i \le m$, $\sum_{t=1}^{T} s_{i,t} = 1$.

- For each subtype, at least one sample is assigned to it: $1 \le t \le T$, $\sum_{i=1}^{m} s_{i,t} \ge 1$.

- Gene $g_j$ is considered mutated in pathway $P_k$ only if it belongs to that pathway: for $1 \le i \le m$, $1 \le j \le n$, and $1 \le k \le K$, $\alpha_{i,j,k} \le p_{j,k}$.

- Each pathway $P_k$ has at most one gene mutated: for $1 \le i \le m$ and $1 \le k \le K$, $\sum_{j=1}^{n} \alpha_{i,j,k} \le 1$.

- For each subtype, pathways are mutated in a linear order, for $1 \le k \le K$ and $1 \le t \le T$, $\sum_{l=1}^{K} (A_{t,k,l} + A_{t,l,k}) = K - 1, (l \ne k)$.

- Each sample satisfies the progression model of its corresponding subtype: for $1 \leq i \leq m$, $1 \leq t \leq T$, and $1 \leq k, l \leq K \, (l \neq k)$, $\sum_{j=1}^{n} \alpha_{i,j,k} C_{i,j} - \sum_{j=1}^{n} \alpha_{i,j,l} C_{i,j} \geq s_{i,t} + A_{t,k,l} - 2 - \epsilon$.

- The values of $\alpha$ is bounded by observed mutations and we assume no false negatives: for $1 \leq i \leq m$ and $1 \leq j \leq n$, $\sum_{k=1}^{K} \alpha_{i,j,k} \leq M_{i,j}$.

The objective function of SPM is as follows:

$$\min \sum_{i,j,k} (C_{i,j} p_{j,k} - C_{i,j} \alpha_{i,j,k}) \qquad (2)$$

In this formula, the term $C_{i,j} p_{j,k}$ represents the observed CCF profiles, and the term $C_{i,j} \alpha_{i,j,k}$ represents the CCF profile "predicted" by the SPM. The objective of SPM is to determine the pathways, subtypes, and the pathway progression models of the subtypes that minimize the difference between the observed CCFs and the CCFs "predicted" by the SPM.

## Conclusions

This work introduces a new mathematical model, called SPM, to simultaneously capture cancer subtypes and pathways as well as the pathway progression models of each subtype. The underlying assumptions are mutual exclusivity of mutations within each pathway, the correlation between CCF value and mutation timing, and the subtype-specificity of cancer progression. Based on experiments with 9 synthetic datasets, SPM is very robust against different problem specifics, such as the number of patients, subtypes and pathways and data noise, while PLPM is sensitive to these factors. Based on experiments with two real datasets, namely GBM and COAD, the detected subtypes correlated to the clinical attributes such as survival and tumor stage. Additionally, the discovered pathways contained genes known to be related to each other. The relative importance of the pathways in different subtypes with respect to the progression models was consistent with the existing knowledge. In some cases, the pathways detected by the SPM were more informative than the pathways identified by an existing model, namely PLPM.

As an NP-Hard problem, similar to PLPM, the ILP solution for SPM is hindered by factors such as quality of the input data and available computational power. This also restricts the number of genes and samples that can be included in the input. Therefore, employing alternative optimization methods is a new direction for future work. Moreover, extending the model to automatically select a subset of the input genes (e.g. based on mutual exclusivity) to be included in the progression inference can increase the generality of SPM. Although the proposed model is applicable to both SNVs and CNVs, this study was restricted to SNVs due to the limitation of current technology for computing CCF values in presence of CNVs. This limitation should be addressed in future work as the required technology becomes available.

## Author Contributions

**Conceptualization:** Sahand Khakabimamaghani.

**Data curation:** Sahand Khakabimamaghani, Dujian Ding.

**Formal analysis:** Sahand Khakabimamaghani, Dujian Ding.

**Investigation:** Martin Ester.

**Methodology:** Sahand Khakabimamaghani, Dujian Ding, Martin Ester.

**Project administration:** Sahand Khakabimamaghani.

**Software:** Dujian Ding.

**Supervision:** Martin Ester.

**Validation:** Sahand Khakabimamaghani, Dujian Ding, Oliver Snow.

**Visualization:** Sahand Khakabimamaghani, Oliver Snow.

**Writing – original draft:** Sahand Khakabimamaghani, Dujian Ding, Oliver Snow, Martin Ester.

**Writing – review & editing:** Oliver Snow, Martin Ester.

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
