## [Decision Letter · Decision Letter 0]

27 Aug 2019

Dear Dr Khakabi,

Thank you very much for submitting your manuscript, 'Uncovering the subtype-specific temporal order of cancer pathway dysregulation', to PLOS Computational Biology. As with all papers submitted to the journal, yours was fully evaluated by the PLOS Computational Biology editorial team, and in this case, by independent peer reviewers. The reviewers appreciated the attention to an important topic but identified some aspects of the manuscript that should be improved.

We would therefore like to ask you to modify the manuscript according to the review recommendations before we can consider your manuscript for acceptance. Your revisions should address the specific points made by each reviewer and we encourage you to respond to particular issues Please note while forming your response, if your article is accepted, you may have the opportunity to make the peer review history publicly available. The record will include editor decision letters (with reviews) and your responses to reviewer comments. If eligible, we will contact you to opt in or out.raised.

- Supporting Information uploaded as separate files, titled 'Dataset', 'Figure', 'Table', 'Text', 'Protocol', 'Audio', or 'Video'.

We hope to receive your revised manuscript within the next 30 days. If you anticipate any delay in its return, we ask that you let us know the expected resubmission date by email at ploscompbiol@plos.org.

Sincerely,

Rachel Karchin

Associate Editor

PLOS Computational Biology

Ruth Nussinov

Editor-in-Chief

PLOS Computational Biology

[LINK]

Reviewer's Responses to Questions

**Comments to the Authors:**

Reviewer #1: Given a cohort of cancer patients with CCFs of each gene, the authors aim to identify (i) a partition of patients into T subtypes, (ii) a partition of genes into K pathways with "mutual exclusivity" for each subtype and (iii) a sequence, or progression, of a subset of pathways for each subtype with decreasing "average" CCFs (using parameter epsilon). The authors show that this problem is NP-hard and introduce an ILP which they run on real data.

I have reviewed the initial submission to RECOMB-CCB. My previous concerns/comments have been satisfactorily addressed. The journal version of the manuscript contains a clearer problem statement, sensitivity analysis using simulated data, and various revisions to clarify exposition, hardness proof and overall presentation of results. I have no further comments. This well-written manuscript will be of interest to the readership of this journal.

Reviewer #2: Building on a previously established Integer Linear program (PLPM) (see ref. [1]), in this study Khakabimamghani et al. have introduced a new method, called SPM to address the reconstruction of the temporal order of pathway dysregulation. They have extended the previous work (see ref. [1]) by using CCF values instead of binarized data and also, they have taken the subtype specificity of cancer progression into account in their formulation of the integer linear program. While their focus on the subtype-specificity of cancer progression is much appreciated, I have major issues with the originality/novelty of their methodology and the solidity of their results. Therefore, although I do not see the potential to recommend this work for publication in PLOS Computational Biology, I am pretty sure that it may warrant publication in an alternative journal (e.g. PLOS One), provided the major concerns are adequately addressed.

Major comments:

Comment #1:

I appreciate the authors attempt to include subtype-specificity of cancer progression in reconstruction of the ordering of cancer pathway dysregulation, which has resulted in some improvements (particularly in the simulation results) as compared to the previous study (see ref [1]). However, I have the impression that their framework is very similar to the previous work and so to my view, it is not sufficiently novel. Moreover, as the authors have clearly acknowledged in the conclusion:

“Due to the complexity of SPM, finding the optimal solution is hindered by factors such as quality of the input data and available computational power. This also restricts the number of genes and samples that can be included in SPM’s input. Therefore, employing alternative optimization methods is a new direction for future work.”

the method seems to suffer from complexity issues that is not well-addressed in this study.

Comment #2:

Although based on their sensitivity analysis using synthetic data, the authors have shown that their method more accurately assigns genes to pathways and more consistently reconstructs their ordering (as summarized in figure 2), their analysis of real dataset, which is exactly the same as the ones used in the previous study (see ref [1]), does not provide conclusive evidence regarding the advantage of including the subtype-specificity of cancer progression. This is acknowledged by the authors in multiple paragraphs in their Real Data Results section:

“Where SPM struggles is in grouping the PI3K pathway genes together, spreading them across identified pathways. This could potentially be due to the solution being sub-optimal as a result of insufficient data and model complexity”

“The separation of genes into their known biological pathways, as defined in [43], is less clear with the COAD dataset, with genes being spread across the identified pathways.”

“There is no clear advantage of any of the four identified progressions compared to the progression from PLMP, as they all struggle to fully recapitulate the true pathways, and are fairly similar, in regards to their identified pathways.”

“The resulting subtypes for COAD failed to show any meaningful difference in the survival analysis, unlike the GBM results.”

Minor comments:

Comment #3:

In the problem definition, the authors mention that: “For loci with copy number variation (CNV), computation of CCF is very difficult and associated with uncertainty. Therefore, we do not consider loci with CNV regardless of occurrence of SNV in the locus”

Would the authors comment on what is the prevalence of loci with CNV? Completely ignoring loci with CNV might have the potential to strongly bias the entire analyses. I therefore recommend the authors to clearly discuss this major issue in the conclusion or discussion section and provide quantitative data to ensure that ignoring loci with CNV data does not strongly mislead the results.

Comment #4:

In page 4, variables tau (Γ) and Z are introduced respectively as: “Γ as the sum of CCFs of genes within pathway x and Z as the corresponding CCF value of pathway x for sample I based on the CCFs of the genes belonging to x (described later) ”. The language is very unclear and according to these definitions, these two variables are almost indistinguishable. I suspect that they are used with different notation in equation (2) in page 12, where apparently the first term (equivalent of tau (Γ)) represents the “observed” CCF profiles and the second term (equivalent of Z) is the CCF profile “predicted” by SPM. I strongly recommend using the same language and notations in both places to avoid major confusions.

Comment #5:

In the real data analyses, the authors have used exactly the same dataset as in (see ref [1]), which is focused only on glioblastoma multiforme and colorectal cancer. To prove the superiority of their method, analyses of alternative real datasets (including different cancer types) are required. At least, analysis of an alternative colorectal cancer dataset (see ref [2]) is necessary, which can help the authors to see whether their conclusions regarding colorectal cancer is robust and reproducible in different datasets?

Comment #6:

In page 8, it is mentioned that “However, based on an empirical test for one of the experiments (not shown) and optimality gaps of ~13% and 0% respectively for COAD nad GBM datasets, the results are expected to be near optimal, if not optimal.”. Please include the (not shown) results at least in the supplementary section.

Comment #7:

Please provide a more informative caption for the figure 4.

Comment #8:

In page 9, it is mentioned that “Additionally, all detected subtypes map well (Rand index =0.68) onto the five subtypes of GBM described in [52].”

It would be helpful, if the authors include a measure of the significance of the Rand index, for example the p-value quantified by cc_test_ari function in CrossClustering R package (see ref [3]).

Comment #9:

In page 9 it is written “… and thus drugs that target early mutations should be more effective”. The logic behind this conclusion is not easy to follow, instead it seems to be highly speculative. I recommend avoiding such sentences at least in the results section.

Comment #10:

In the main text, Figure 4b is discussed after figure 5. The reordering of the figures accordingly may be required.

Comment #11:

In the conclusions section in page 12, the authors conclude that

“Based on experiments with two real datasets, namely GBM and COAD, the detected subtypes correlated to the clinical attributes such as survival and tumor stage.”

I realized that this conclusion is untrue as it contradicts the author’s own results, which was previously described as:

“The resulting subtypes for COAD failed to show any meaningful difference in the survival analysis, unlike the GBM results.”.

Comment #12:

In the conclusions section in page 12, the authors conclude that “The relative importance of the pathways in different subtypes with respect to the progression models was consistent with the existing knowledge.”

I realized that this conclusion is also untrue as it contradicts the author’s own results, which was previously described as:

“The separation of genes into their known biological pathways, as defined in [43], is less clear with the COAD dataset, with genes being spread across the identified pathways.”

“There is no clear advantage of any of the four identified progressions compared to the progression from PLMP, as they all struggle to fully recapitulate the true pathways, and are fairly similar, in regards to their identified pathways.”

Comment #13:

The last sentence of the conclusions in page 13 is too general and to my view is far beyond the scope of what this work has achieved: “The proposed framework provides a basis for gaining insight into the biology of cancer and its treatment as sequencing technologies advance, the quality of data increases, and larger datasets become available”.

References:

[1] Raphael BJ, Vandin F. Simulatenous inference of Cancer Pathways and Tumor Progression from Cross-Sectional Mutation Data. In: Sharan R, editor. Research in Computational Molecular Biology. Cham: Springer International Publishing; 2014. P. 250-264.

[2] Wood L.D., Parsons D.W., Jones S., et al. 2007. The genomic landscapes of human breast and colorectal cancers. Science 318, 1108–1113.

[3] E.M. Qannari, P. Courcoux and Faye P. (2014) Significance test of the adjusted Rand index. Appli-cation to the free sorting task, Food Quality and Preference, (32)93-97

**Have all data underlying the figures and results presented in the manuscript been provided?**

Reviewer #1: Yes

Reviewer #2: Yes

PLOS authors have the option to publish the peer review history of their article (what does this mean?). If published, this will include your full peer review and any attached files.

Reviewer #1: Yes: Mohammed El-Kebir

Reviewer #2: No

---

## [Editor Report · Decision Letter 1]

30 Sep 2019

Dear Dr Khakabimamaghani,

We are pleased to inform you that your manuscript 'Uncovering the subtype-specific temporal order of cancer pathway dysregulation' has been provisionally accepted for publication in PLOS Computational Biology.

In the meantime, please log into Editorial Manager at https://www.editorialmanager.com/pcompbiol/, click the "Update My Information" link at the top of the page, and update your user information to ensure an efficient production and billing process.

One of the goals of PLOS is to make science accessible to educators and the public. PLOS staff issue occasional press releases and make early versions of PLOS Computational Biology articles available to science writers and journalists. PLOS staff also collaborate with Communication and Public Information Offices and would be happy to work with the relevant people at your institution or funding agency. If your institution or funding agency is interested in promoting your findings, please ask them to coordinate their releases with PLOS (contact ploscompbiol@plos.org).

Thank you again for supporting Open Access publishing. We look forward to publishing your paper in PLOS Computational Biology.

Sincerely,

Rachel Karchin

Associate Editor

PLOS Computational Biology

Ruth Nussinov

Editor-in-Chief

PLOS Computational Biology

---

## [Editor Report · Acceptance letter]

31 Oct 2019

PCOMPBIOL-D-19-00937R1 

Uncovering the subtype-specific temporal order of cancer pathway dysregulation

Dear Dr Khakabimamaghani,

I am pleased to inform you that your manuscript has been formally accepted for publication in PLOS Computational Biology. Your manuscript is now with our production department and you will be notified of the publication date in due course.

With kind regards,

Matt Lyles
